# The Urban CoCreation Lab—An Integrated Platform for Remote and Simultaneous Collaborative Urban Planning and Design through Web-Based Desktop 3D Modeling, Head-Mounted Virtual Reality and Mobile Augmented Reality: Prototyping a Minimum Viable Product and Developing Specifications for a Minimum Marketable Product

**Hyekyung Imottesjo** [1],* and **Jaan-Henrik Kain** [2]

1   Department of Architecture and Civil Engineering, Chalmers University of Technology, 412 96 Gothenburg, Sweden
2   Gothenburg Research Institute, GRI, University of Gothenburg, 405 30 Göteborg, Sweden; jaan-henrik.kain@gu.se
*   Correspondence: kyung@chalmers.se; Tel.: +46-73-9972-915

**Abstract:** Both policy and research highlight the importance of diverse stakeholder input in urban development processes but visualizing future built environments and creating two-way design communication for non-expert stakeholders are challenging. The present study develops an intuitive and simplified 3D modeling platform that integrates web-based desktop, virtual reality and mobile augmented reality technologies for remote simultaneous urban design collaboration. Through iterative prototyping, based on two series of workshops with stakeholders, the study resulted in such an integrated platform as a minimum viable product as well as specifications for a minimum marketable product to be used in real projects. Further study is required to evaluate the minimum level of detail in the 3D modeling necessary for good perception of scale and environmental impact simulation.

**Keywords:** collaborative urban designing; web-based desktop 3D modeling; mobile augmented reality; virtual reality; minimum viable product; minimum marketable product; integrated 3D modeling platform

## 1. Introduction

The importance of encouraging a diversity of stakeholders to be engaged in urban planning and design through participation and dialogues in all phases of development of the built environment—envisioning, designing, construction and maintenance—is stressed in both policy [1–3] and research [4,5]. Urban planning and design involve complex challenges that require a wide range of solutions, a diversity best supported through the engagement of many different stakeholders [6–10] supplying valuable knowledge developed through the complex relationship between local stakeholders and urban landscapes [11]. This is especially evident during this time of significant global issues, including climate challenges [12], financial crisis [13], and, more recently, the global pandemic [14]. The complexity of these challenges has become evident, especially regarding the need for continuous adaptation in the face of unpredictability, where the participation of various stakeholders in the urban development process is essential for urban resilience [2,15]. Continuous dialogue between the stakeholders through top-down and bottom-up channels [16] is required to support such adaptivity capacity [6,17].

In this context, a main challenge in stakeholder participation in urban planning and development processes is representing and communicating unbuilt urban planning and design proposals [18,19]. Efficient and unbiased communication of 3D data necessitates new

types of platforms for stakeholder participation [20,21]. This has been both demonstrated and exacerbated by the COVID-19 pandemic, additionally charging the challenges of facilitating remote working and collaboration [22,23].

This article will present one type of response to this challenge of broad stakeholder participation in urban planning and design, based on web-based 3D modeling, virtual reality, and mobile augmented reality.

### 1.1. Challenges of Stakeholder Engagement in Urban Planning and Design

#### 1.1.1. Representation of the Built Environment

Transparency of information and data provides knowledgeability to the stakeholders [24–26]. Such knowledgeability increases the quality of stakeholder participation by providing both relevance and validity of the outcome of the stakeholder engagement [21,27]. In particular, understanding how urban development plans and designs would affect the future built environment is fundamental. Here, various visualization techniques are used to communicate the visions, designs and consequences of urban plans and designs to stakeholders. Among these techniques, traditional tools for visualization, such as 2D drawings of plans, sections and elevations, or 3D rendered images of building volumes, have shown to be lacking in communicating the complexity of urban spatial information to non-expert stakeholders [18,28–32], thus hindering those stakeholders from creating valid responses.

As a response to these shortcomings, different types of immersive mixed-media applications have gradually surfaced as visualization technologies, with the potential to better communicate urban built environment information and data to the public, including virtual reality (VR) and augmented reality (AR) in conjunction with 3D modeling software [3,20,29,33–35] with the potential to provide multi-modal location-based services [34]. Additionally, information and communications technologies (ICT) have developed significantly, and there is widespread availability of ICT tools, such as smartphones and head-mounted VR and AR visors. Coupled with increased connectivity and capacity of such devices owned by the public, this makes it possible to introduce and investigate the use of immersive technologies for the inclusion of a diversity of stakeholders in urban planning and design discussions [34,36–38]. Mobile augmented reality (MAR) is one such technology, enabling stakeholders to experience an immersive presentation of the future built environment through their smartphones, providing AR accessibility, portability and mobility. Furthermore, the current necessities for remote access to information and data through visualization and representation due to the pandemic and its ensuing physical limitations for collaboration bring a new dimension to our interest in mixed-reality solutions for remote communication and collaboration [39].

#### 1.1.2. Facilitating Remote Two-Way Communication

Mixed-reality visualization technologies for stakeholder communication have been studied and, to some extent, been used in the disciplines of architecture and urban planning and design [20,40]. Previous research indicates that the use of AR in conjunction with the use of mobile phones for citizen evaluation of built environment proposals bring enhanced feedback from citizens compared to the use of other forms of visualization and representation [37,41–43]. Likewise, immersive VR and using 3D modeling platforms for urban environment representation have shown an increased quality of citizen engagement [44,45]. In architecture, VR has been used for communication between the stakeholders in actual projects [46,47], showing cost benefits and supporting evaluations based on increased spatial awareness by stakeholders even though the implemented modification is only made virtually [40].

Furthermore, to facilitate two-way communication between the urban planners and designers and the stakeholders, there is a need for possibilities for non-expert stakeholders to put forward their vision [37,48–51]. Here, non-expert stakeholders include those who are not familiar with architectural and urban planning visualization methods, including the use of 2D and 3D drawing and modeling software. They would typically have difficulties

easily understanding and interpreting architectural drawings and urban plans and lack the skills to fully bring forward their vision of spatial design using the digital visualization medium.

While immersive technologies seem to provide stakeholders with increased spatial perception [37,38,41,51,52] and heightened engagement [45,50,51], stakeholder input is still often created through verbal or written feedback [50]. Some 3D-modeling software and VR platforms directly reflect user feedback by modifying the spatial design [40,45,53]—changes experienced by all stakeholder simultaneously. Still, in these cases, a designer or a 3D modeler is present at the sessions with the stakeholders, modifying the spatial design using 3D-modeling software based on the stakeholder feedback. The stakeholders can then experience the changes while being immersed in the VR environment. For example, if the width of a corridor is experienced to be too narrow by a stakeholder, the 3D modeler increases the width, and the impact of the stakeholder input is directly experienced. This has shown an increased efficiency in communication between the designers and end-users, as well as cost benefits in design communication and modification [40,54].

Multiple-user collaboration in shared virtual environments—especially when combined with immersive visualization technologies—has the potential to facilitate simultaneous design collaborations where stakeholders can collectively plan, design and create virtual environments [55,56]. Such multiple user virtual environments (MUVEs) allow users "to access virtual contexts, to interact with digital artifacts, to represent themselves through 'avatars,' (and) to communicate with other participants" [57]. From the planning and design collaboration perspective, MUVEs have been seen to support remote collaboration and education in architectural designing [56,58,59]. Moreover, collaborating while observing and visualizing other stakeholders' movements and design trials enhanced explorative creativity compared to working alone [59]. Desktop-based MUVEs, where diverse stakeholders collaborate and communicate through VR and AR media, might, thus, provide efficient communication and collaboration platforms in the context of complex spatial issues related to the urban environment [59,60]. Especially bringing mobile (e.g., smartphone-based) AR (MAR) into such MUVEs would facilitate taking the task of designing built objects away from the 3D virtual environment out to the actual project site [37], with benefits of providing the site-specific "auditory, olfactory, haptic and kinetic experiences" [28] inherent in urban spaces.

### 1.1.3. Knowledge Gap and Aim

In the examples of two-way communication described above, the design modification that the stakeholders responded with and experienced in virtual immersion required a visualization expert who interprets the feedback through 3D modeling on site. Nevertheless, if the aim is to strengthen two-way communication with layperson stakeholders, how can stakeholders with varying levels of expertise using the modeling software required for such visualization create feedback through visual input autonomously, without involving a 3D modeler at every step? Setting up and executing own agendas by creating multiple ways of communication without having expert knowledge in software development has been the strength of WEB 2.0 [61]. Social media platforms and multiple user collaboration platforms, such as Google Drive, are examples of such self-organized communication channels. However, there is still a lack of MUVEs where non-expert stakeholders can develop collaborative projects based on immersive visualization and representation technologies—without any middleman experts needed—to co-create their visions of the future environment [38,41,47,50,62].

This article presents a research and development project that aims to develop a platform for remote and simultaneous stakeholder collaboration in urban planning and design, integrating web-based desktop 3D modeling, VR and MAR. The project intends to offer better accessibility to such tools to diverse stakeholders without prior knowledge in 3D modeling and drawing. By enabling the use of different types of devices, such as laptops, VR goggles or mobile phones, stakeholders should be able to create and present

visual input either at home, while using web-based 3D desktop stations or VR devices at a dedicated VR facility, or on-site using smartphones. The first two phases of this ongoing project study are presented in this article. Phase 1 developed the first set of platform specifications leading to a minimum viable product (MVP) with an integrated platform for design collaboration using a web-based desktop station, VR and MAR. An MVP is an early, basic version of a product (such as a piece of technology and a computer program) that meets the minimum necessary requirements for use but can be adapted and improved in the future, especially after customer feedback [63]. In this case, the MVP was developed by testing the rudimentary prototype, integrating MAR, VR and web-based desktop 3D modeling stations for simultaneous synchronization of the devices. Only the very basic user functions were included, i.e., placing a 3D cube as a building volume, modifying this cube by moving, rotating, scaling, deleting and navigating in the virtual space, and communicating with other stakeholders. Based on phase 1 and its MVP, the next phase developed and sharpened the specifications further, delivering a concise starting point for the development of a minimum marketable product (MMP), i.e., a platform fulfilling the minimum requirements for the platform to be launched in the market as a usable tool with possibilities to be integrated urban planning and design processes as a communication platform.

## 2. Methods

The Urban CoCreation Lab is a collaborative science–practice project involving Akademiska Hus (Gothenburg, Sweden) (a real estate company for universities owned by the Swedish state), Johanneberg Science Park (a collaborative arena for urban development), Lindholmen Visual Arena (a neutral platform promoting new ways of visualizing), Atvis AB (Gothenburg, Sweden) (a software and visualization company), Gothenburg Research Institute and Chalmers University of Technology. Akademiska Hus provided the project with two ongoing campus development projects in Gothenburg, Sweden, as cases. Through the collaboration, it was possible to develop the urban co-creation platform through testing in real urban development cases driven by the users and their experience.

The platform was developed through iterative prototyping to explore various alternatives and work out details (i.e., increasing precision) [64]. Iterative prototyping was explicitly chosen as a suitable development method for working within conditions of limited budget and time [65,66], combined with uncertainties related to the final costs and results of software development [67,68]. Iterations were made during two series of workshops, one for the MVP (Phase 1) and one for the MMP (Phase 2) (see Figure 1). In each phase, the iterations were made in two steps. The first step consisted of workshops where various stakeholders tried out a rudimentary prototype and, based on these trials, contributed with a wishlist of what features are needed for a platform to function as a remote, simultaneous collaborative design tool. The wishlist was compiled by taking notes, based on semi-structured group discussions and interviews, and individual questionnaires (see Supplied materials). In the second step, the wishlist was turned into detailed specifications by software developers and researchers based on priority ranking and implementability of the wished features, and a next-generation platform was developed to be explored in the following workshop. This process of bottom-up identification of necessary features through a series of user-test workshops based on actual urban development cases was intended to provide specifications that neither software designers nor researchers could directly know, and which stakeholders cannot explicitly explain prior to actual hands-on application in real scenarios [69].

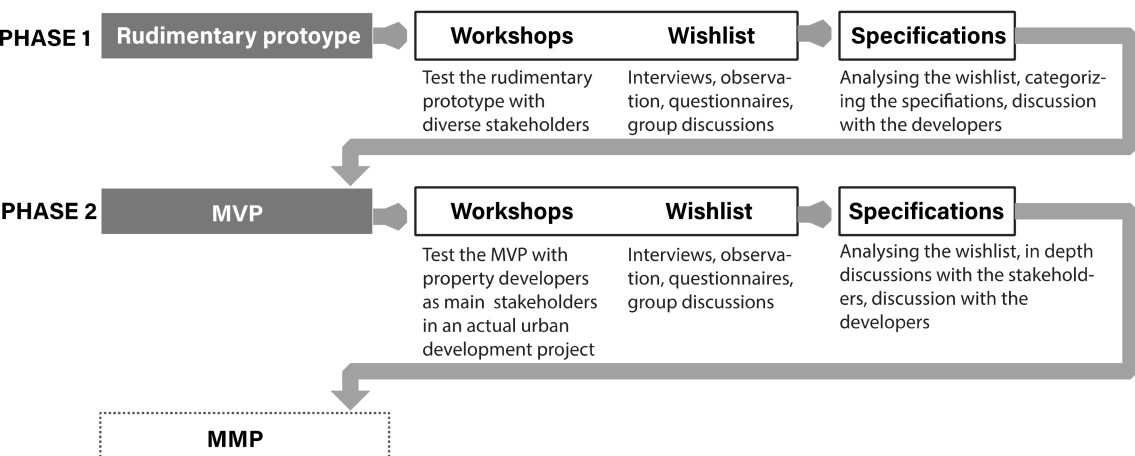

**Figure 1.** The Urban CoCreation Lab development process, Phases 1 and 2.

*2.1. Phase 1: Developing a Minimum Viable Product*

2.1.1. Case Gibraltarvallen

The Gibraltarvallen campus project concerns 130,000 m$^2$ of total floor area for mixed-use development. The detailed plan for the area is in an advanced stage and in the process of being legally adopted. The project would hence belong to the 6th phase of Akademiska Hus' property development scheme—"Implementation" (see Figure 2)—already having passed through the first five phases of "Starting up," "Analysis," "Vision," "Design," and "Anchoring". Although designs will undergo fine-tuning to comply with the comprehensive vision of the project area, the design process is mostly finished at this stage. The current implementation focuses on developing housing on a small part of the site, involving two property developer firms and two architecture firms that have made the design proposals. As the whole project area is quite large, there is a need to discuss how those areas, that will remain unbuilt for a long time, can be used until ultimately being designed and built in the future.

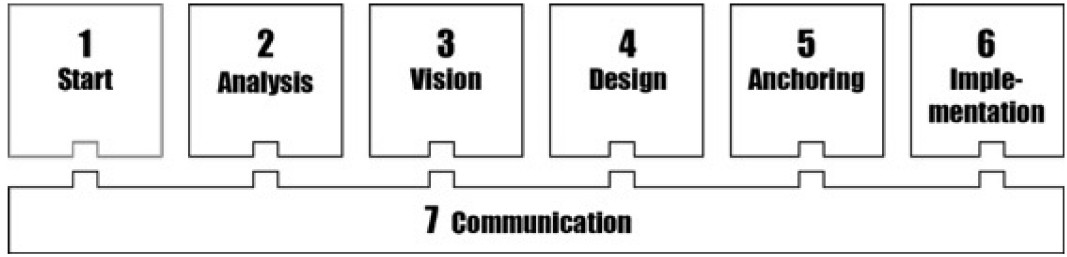

**Figure 2.** The campus development process stages developed by Akademiska Hus.

2.1.2. First Prototype Trials and Gathering Stakeholder Wishes

Phase 1 comprised three workshops with three types of participating stakeholders, categorized according to the type of involvement in the campus development project: 1. internal (Akademiska Hus) property developer stakeholders, 2. internal stakeholders from involved architect firms and 3. external stakeholders affected as users of the space, in this case, university students and teachers, all in all, eight participants. The objective of this first series of workshops was to gather a gross list of as many types of stakeholder wishes as possible. The workshops were recorded to facilitate detailed analysis. Due to the COVID-19 pandemic, the number of participants had to be limited. After each participant had tested the VR goggles or smartphones, the equipment had to be sterilized for subsequent use. The use of masks and hand sanitizers was encouraged. It can be assumed that these measures affected the user experience to some degree.

These workshops tested a first rudimentary prototype platform developed from a set of tentative specifications formulated by the researchers and software developers. This tentative platform integrated and enabled simultaneous synchronization between three visualization mediums: web-based desktop-based 3D modeling, VR goggles and MAR smartphones with a minimum delay (<5 s) between the devices (see Figure 3). To gather as many wishes as possible for developing the minimum requirements for the platform, only rudimentary functionalities were tested. The first part of the workshops tested 3D model projection through head-mounted VR and MAR, and the latter in various scales, such as 1:400 (see Figure 3). The second part of the workshops took the participants out to the project site to try out the 1:1 scale on-site MAR.

During the first part of the workshops, participants were (a) shown how the devices work, (b) divided, so that each participant tried one device at a time, (c) asked to try navigating in the virtual environment, (d) asked to place a 3D cube as a building volume, (e) asked to change the scale, location and rotation of the building volume and (f) asked to communicate with the other participants to change the building volume (see Figure 4). This part of the workshops was then concluded by joint discussions (recorded) directed at capturing stakeholders' wishes for platform development.

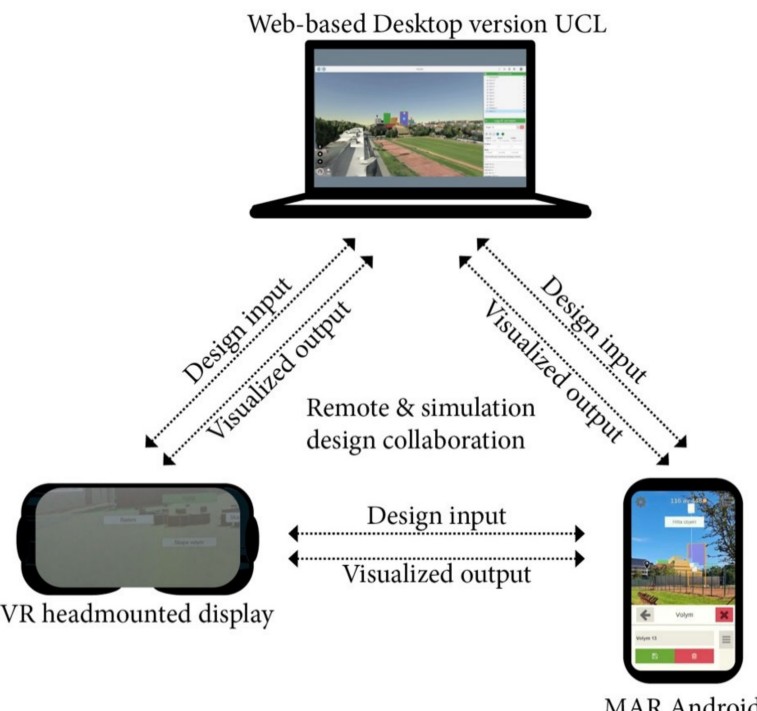

**Figure 3.** Synchronizing web-based desktop stations, VR and MAR.

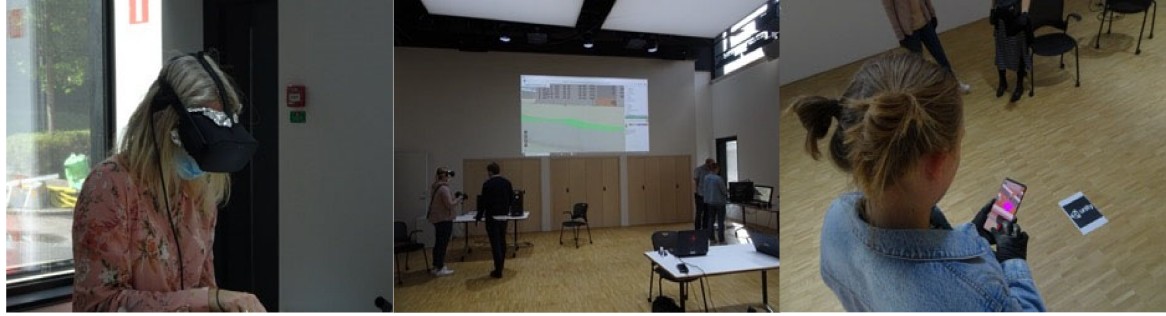

**Figure 4.** Stakeholder communication at workshops in Phase 1.

The next step of the workshops was for the participants to walk to the project site and view the 3D building volumes created in the first part of the workshops, but now through smartphone-based MAR on site and in scale 1:1 (see Figure 5). They were also asked to change the volumes by scaling, rotating, or moving them in the AR environment. Again, this on-site exercise was concluded by groups discussions (recorded) to capture stakeholder wishes.

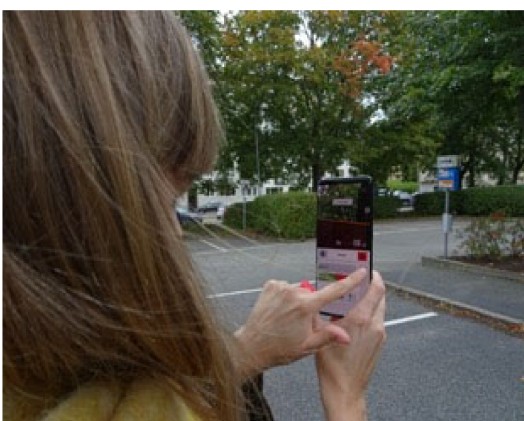

**Figure 5.** Stakeholder testing 1:1 on-site MAR.

The final step of the workshops was to walk back indoors to have some refreshments, carry out interviews and group discussions (recorded), and ask the participants to fill in the questionnaires.

### 2.1.3. Developing Specifications and Shaping a Minimum Viable Product

A consolidated wishlist was developed based on input from all three workshops. Participants' discussions, interviews and questionnaires were transcribed. The transcription was then analyzed to rank the frequency of the words used by the participants, for example, "scale" or "floor". These ranked terms were analyzed further according to the context within which they were used. For example, the term "floor" was mentioned both in the context of "visualizing the details" and in "perception of building scale". Based on this wishlist analysis, specifications were developed through thematic clustering to create broad themes and sub-themes that correspond to the terminologies frequently mentioned and to the context in which they were mentioned. For example, wishes linked to possibilities to distinguish between existing and proposed buildings were fused into the specification "different models/building projects to be put in different layers so that they can be turned on and off to show," located under the broad theme "visualization" and the sub-theme "layer".

After the thematic clustering, the specifications were ranked according to how frequently the participants demanded them. The specifications were then brought to the software developers to fine-tune and assess their implementability. Finally, the highest-ranking and most implementable specifications for each sub-theme were chosen to be developed in the MPV. For example, "compatibility with other platforms", such as other existing CAD software formats with file types frequently used, such as the dwg-file type, was not selected due to challenging license matters even if technical solutions could have been implemented. Once the final selection of specifications for the MVP was established, the software developers implemented measures in the MVP to comply with the specifications as closely as possible within the available resources in time and funding.

### 2.2. Phase 2: Developing a Minimum Marketable Product (MMP)

#### 2.2.1. Case Medicinarberget

For the second phase, another real-case campus development project was used as a case. This project is in the 3rd phase of campus development, defined as the "Vision"

phase (see Figure 2). In this phase, a broad vision of the urban area is being developed with consideration to its long-term development and the impact of the development on the city, as a whole. Among other university facilities, the project includes developing 800–1000 student housing units, and a new detailed plan is underway. The detailed planning consists, e.g., of the projected number of housing units, green area development, proportions of mixed functions, solutions for parking and proposals for flexible use of buildings. The heights of the buildings and how to develop the connections to surrounding urban areas were also to be discussed.

### 2.2.2. Trials of Minimum Viable Product and Gathering Additional Stakeholder Wishes

In this phase, due to increased restrictions and limitations due to COVID-19, the research focused on developing the remote simultaneous web-based desktop version of the platform to limit physical meetings and gatherings during the workshops, thus carried out as Zoom meetings. Since the VR and MAR systems are integrated into the desktop version, modifications on the desktop version are also applied to the other interfaces. This means that, although COVID-19 reduced the possibilities to fully develop the integration of VR and MAR based on stakeholder input, the MAR system was continuously revised according to the proposed changes, and its integration with the Web-based desktop version was tested continuously by the researchers and software developers.

Again, three stakeholder workshops were carried out in Phase 2, but now with only two internal stakeholders as participants, i.e., the property developers from Akademiska Hus. The objective was to test the MVP to identify remaining issues and further needs by once more applying the platform in a real development process—but in another phase of campus development—to develop specifications for an MMP.

During the first two workshops of Phase 2, the participants explored the 3D model of campus development project using the MVP platform remotely through their web-based desktops/laptops. Two researchers and a software developer were present for observation and note-taking and to solve any issues linked to accessing and handling the platform. The sessions were carried out as Zoom meetings with screen sharing and were recorded (see Figure 6). With occasional help and guidance from the software developer, the two participating stakeholders had ample opportunities to test all the platform's features in relation to developing the campus area and evaluate different development strategies. Without input from the software engineers, the stakeholders began to emulate the property development process. This led them to test re-creating road networks, demolishing existing buildings, adjusting terrain heights, placing and modifying building volumes, pinning interesting points and perspective points and following the other participant's point of view to present and discuss perspectives of property development with each other. The participating property developers also placed MAR markers at the points of interest to be used for the MAR on site. The two workshops were concluded by carrying out discussions and by stakeholders filling in the questionnaires to capture further wishes linked to encountered issues and further needs.

Based on the observations, discussions and questionnaires, a new wishlist was added to the specifications previously developed for the MVP. For example, a wish for "different textures for the buildings including transparency" was added under the broad theme "visualization" and the sub-theme "detailing".

The third workshop of Phase 2 was designed to make the wishlist from the previous two workshops more detailed with the objective to develop final specifications for the MMP. As a preparation for this final workshop, the material gathered in the first two workshops of Phase 2 was analyzed and articulated into a list of items that needed specific details. For example, a wish for "add building functions" was turned into questions such as: Which building functions should be included for this project? How to develop a user-defined building function? Moreover, some of the wishes identified already in the workshops in Phase 1, and repeated in Phase 2 were investigated further. For example, the issue of "perception of scale" was included in the wishlist from Phase 1. It was dealt with in

the MVP by indicating the floor numbers when modeling but was still experienced as a problem in the workshops in Phase 2.

During this third workshop, the two stakeholders once again went through the campus development steps using all the functions available in the platform: preparing the ground, developing roads, placing 3D cubes to make building volumes, adding pinpoints of interest, and placing MAR markers. During this work process, the researchers discussed the wishes from the articulated list to identify more exact needs. For example, the wish from a previous workshop—"more forms and shapes need to be added"—was discussed with the stakeholders for more details of the actual needs for more forms and shapes. The purpose of going through the whole process was to cover the detailed specifications needed to make the platform fully usable and possible to incorporate in an ordinary work process of Akademiska Hus during their property development stage of "vision".

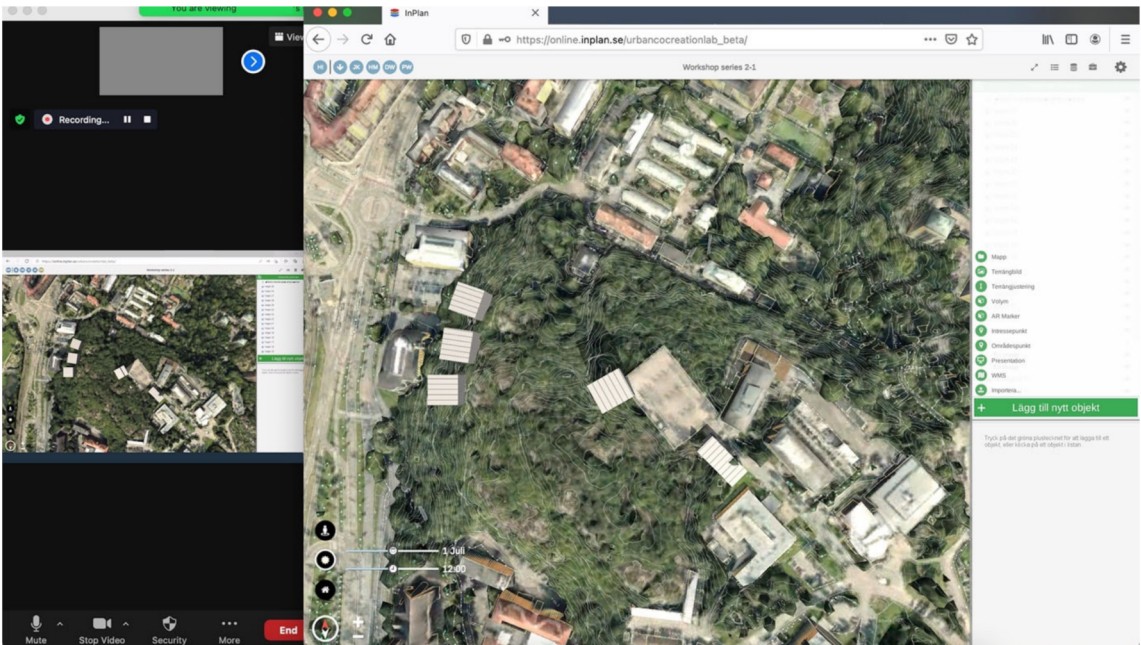

**Figure 6.** Zoom session with the screen sharing function. The image has been modified to protect personal information. The green buttons on the right indicate (from the top): Folder, Terrain image, Terrain adjustment, Volume, AR marker, Interest point, Area point, Presentation, WMS, Import. The plus button is used for placing a new object.

## 3. Results

Below, the sequence of results deriving from the iterative prototyping is described. The section is divided into five parts. First, the basic rudimentary prototype platform used as a starting point for MVP development is presented. Thereafter, the stakeholder wishlist and the resulting specifications resulting from the three workshops in Phase 1 are described. The third part presents how the specifications were implemented to improve existing functions and create one new function, resulting in the MVP. Part four describes the detailed specifications from the workshops in Phase 2, providing the foundation for developing the MMP.

### 3.1. The First Basic Platform Rudimentary Prototype

The first rudimentary prototype was formulated by the researchers and software developers by combining a MAR application previously developed by the researchers (Urban CoBuilder, see [37,70]) with a web-based desktop and head-mounted VR 3D modeling platform previously developed by the software engineers (Inplan, see [71]).

The functions of the rudimentary prototype included:

1.  Synchronization of MAR, web-based desktop 3D modeling stations and head-mounted VR for remote and simultaneous 3D-modeling collaboration;
2.  3D modeling in MAR, web-based desktop stations and head-mounted VR by placing a cube as a building volume which can be modified by rotating, scaling, moving, changing colors and deleting;
3.  Modification of all 3D volumes in MAR, web-based desktop stations and head-mounted VR by all participants, including those created by other participants;
4.  All participants can view the others as simple avatars;
5.  MAR using marker-based localization;
6.  MAR can be viewed as scaled and in scale 1:1;
7.  Participants can navigate in the virtual space and return to a pre-set starting point;
8.  Participants can place a pin in an area of interest and leave a comment, image or sound.

### 3.2. Stakeholder Wishlists, Platform Specifications and Implementation

The workshops resulted in developing four broad themes and 14 sub-themes based on grouping the wishes in the wish list (see Table 1). The specifications were developed for each sub-theme and finally implemented by the software engineers.

What has been finally implemented to comply with the specifications is included in the table. Some of the specifications were not implemented; however, some of the solutions could approximate the effect of implemented specifications. Ex) "Build by adding floors" was implemented by improving how the building volume is scaled up and down in conjunction with adding the floor lines on the 3D texture (See Figure 7).

The implementations of specifications improved the existing functions and user-interface, and in some cases, added as a new function. If an implementation improves a specific function, the related function is listed in the table below.

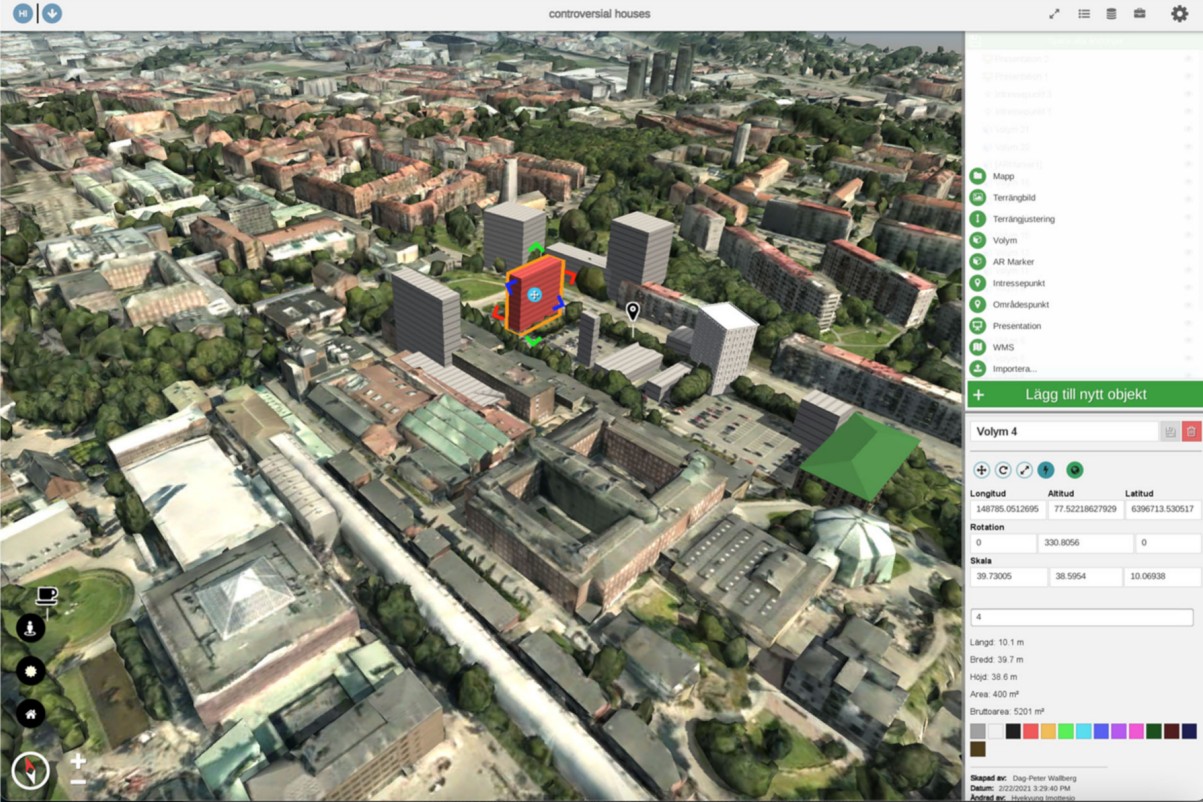

**Figure 7.** The volume function when placing objects, allowing users to place 3D cubes which can be manipulated.

**Table 1.** Stakeholder wishlist, specifications, implementations and related functions for the MVP.

| **Theme 1: Visualization** | | | |
|---|---|---|---|
| **Sub-Theme 1-1: Layers** | | | |
| *Wishes* | *Specifications* | *Implementations* | *Functions* |
| Turn on and off different 3D volumes and models. | Different volumes, models, projects, and information should be placed in layers and turned on and off to enable and disable showing. | A folder structure where users can put models in different folders which can be turned on or off. | Place objects: folders. |
| Visualize different types of information, e.g., underground and water drainage. | | | |
| Add layers with more detailed visuals for specific areas. | Make it possible to highlight certain focus areas. | Highlighting focus objects and areas was not implemented for the MVP. However, more detailed visuals for focus areas can be added, e.g., by inserting 3D drone images. | |
| Clearly separate newly designed and yet-unbuilt projects from existing areas. | | | |
| **Sub-Theme 1-2: Scale** | | | |
| References to understand scale of the building volumes. | Number and height of floors should be indicated. | The facade texture has lines indicating floors. Users can set the height of the floor. | Place objects: Volumes. |
| Indicate how tall a building is. | | | |
| References to understand human scale and terrain. | Reference objects such as a scaled human should be included. | This was not implemented for the MVP. | |
| **Sub-Theme 1-3: Information** | | | |
| Tag buildings with information, such as owners, size, floors and functions. | Tag the building volumes with building information: functions of the building, owner (who created it), floor area and number of floors. Create ability to add comments to buildings. | Floor areas, number of floors and heights are indicated. | Place objects: Volume. |
| Link text/images to objects. | | User can tag an area or object with a pin with information using text, images and sound. | Place objects: Interest area and interest object. |
| **Sub-Theme 1-4: Detailing** | | | |
| Better details for the ground and terrain, such as site boundaries and road edges. | Site boundaries/edges of the streets and roads need to be clearly indicated. | Users can import e.g., a detailed plan as an image file overlay. This image can be modified by rotating, scaling and moving. | Terrain drawing: Import image. |
| | | User can draw lines directly on the terrain in different line weight and color. | Terrain drawing |
| More details in the model, such as facade, doors, windows and roofs. | This was not turned into a specification for the MVP but was still considered important. | | |

**Table 1.** *Cont.*

| *Sub-Theme 1-5: Orientating and locating* | | | |
|---|---|---|---|
| 2D map/terrain image for better orientating. | A 2D map view should be enabled. | Users can choose a layer of plans, maps or satellite images to be overlaid. | Map layers |
| Teleporting to a saved view-point. | Teleporting to a saved view should be enabled. | Users can quickly move back to a saved location using a home button. | A home button |

**Theme 2: Modeling**

*Sub-Theme 2-1: Methods*

| *Wishes* | *Specifications* | *Implementations* | *Functions* |
|---|---|---|---|
| Set width and height of objects when designing 3D volumes. | Building the 3D model by adding the number of floors should be possible. | Building by adding floors was the most mentioned building method in the wishlist. Even though this has not been implemented, adding a facade texture indicating floor lines and being able to modify the height of each floor was implemented. | Place objects: volumes. |
| Set floor number and height when designing 3D volumes. | | | |
| Building volumes by adding floors. | | | |
| Draw polylines and extrude into volumes. | These were not made into specifications for the MVP. | | |
| Build volume by drawing walls. | | | |

*Sub-Theme 2-2: Viewports*

| | | | |
|---|---|---|---|
| Street-view function. | Easily switch mode of view between street-view and bird's-eye view. | This was not implemented for VR and MAR but in the web-based desktop version it is possible to navigate in both street-view and bird's-eye view through a view mode switch button. | A view mode switch button. |
| Switch between 3D immersion and bird's-eye view. | | | |
| Move down to ground and zoom. | | | |

*Sub-Theme 2-3: Functionalities*

| | | | |
|---|---|---|---|
| Save different iterations, histories, geometries and specific views. | Create a model and save it and go back to a previous version. | Building volumes can be saved as presentation viewpoints or recorded views and can be turned on and off for showing. | Place an object: Presentation. |
| User defined names and colors for building volumes. | Users can define names and colors for the building volumes. | Users can modify the color of the building volumes and name them. | Place an object: Volume. |
| Undo/regret and restore. | This was not turned into a specification for the MVP. | | |

**Table 1.** *Cont.*

| | | | |
|---|---|---|---|
| *Sub-Theme 2-4: User Interface* | | | |
| More intuitive and simpler steps to make building volumes. | More simplistic input for basic functions. | User interface was generally improved. | |
| More graphics, less text. | | | |
| **Theme 3: Remote simultaneous collaboration** | | | |
| *Sub-Theme 3-1: Remote collaboration* | | | |
| Guided tour function with preset tracks with limitation to deviate from paths or use of arrows to guide. | Showing other users focus areas by guiding, limited views and possibility to teleport them. | This was implemented as a presentation mode. The users can also adopt the point of view of each other by simply clicking on the other participant's name. Another feature is the adoption of an avatar as a camera, which shows the location of other users as well as where the users are looking. AR markers can be placed by the users for communication across the devices. | Place an object: Presentation. |
| | | | Adopt another user's view. |
| | | | Place an object: AR marker. |
| Add possibilities for feedback and comments. | Create ability to add comments to buildings. | User can tag an area or object with a pin containing information by using text, images, and sound. | Place objects: Interest area and interest object. |
| Highlight focus areas. | This was not turned into a specification for the MVP. | | |
| *Sub-Theme 3-2: Compatibility with other platforms* | | | |
| Compatibility with other tools which architects use, such as Autodesk, 3D-modeling software and Esri. | Import and export models from/to other popular architectural tools. | Sketchup models in compressed FBX format and image files in PNG format can be imported. | Place an object: Import. |
| Connect with Google maps and Sketchup. | | | Terrain drawing: Import image. |
| *Sub-Theme 3-3: Isolation and synchronization* | | | |
| Define level of users for different functionalities, such as only commenting, only viewing and modifying designs. | Users' levels should be defined. | Guest access with only viewing. | |
| Locking and isolating feature for input from individual users so that other users are not allowed to modify. | Possibilities to lock a volume or model should be investigated. However, as the purpose of this tool is collaboration and gathering valuable input from diverse stakeholders, this needs careful consideration how it can best be done. At this stage, this was not turned into a specification for the MVP. | | |
| Ownership of the building volumes and designs. | This was not turned into a specification for the MVP. | | |

**Table 1.** *Cont.*

| | | | |
|---|---|---|---|
| **Theme 4: Simulation** | | | |
| *Sub-Theme 4-1: Design consequences* | | | |
| Sun and shadow simulations. | Impact of shadows based on the design should be simulated. | Sun and shadows can be simulated based on the time and date and the location of the site. | Tools: Sun and shadow simulation. |
| Visualization of environmental qualities, such as air qualities and noise levels. | This was not turned into a specification for the MVP but was still considered relevant. | | |
| *Sub-Theme 4-2: Preset scenarios* | | | |
| Building heights limitations. | Preset scenarios should be simulated. | This was not implemented as a separate function. However, different scenarios can be placed in folders to enable and disable viewing. | Place an object: Folder. |
| Building orientations. | | | |

### 3.3. Implementation into a Minimum Viable Product

The Urban CoCreation Lab MVP is a web-based 3D modeling platform with a number of functions, based on 3D mesh models of cities and integrated with MAR and VR (see Table 2). Participants can freely move around the virtual environment, both in fly-over mode and walking through the streets as avatars. Participants can view each other as simplified avatars in the virtual environment as well as in MAR (see Figure 8). They can also take on other participants' perspectives and follow what others are looking at and working on. It is possible to change the 3D terrain by adjusting the heights of the terrain (see Figure 9), including removing existing buildings. Lines can be drawn directly on the terrain (see Figure 10), and 3D cubes can be placed as building volumes, which the participants can then manipulate by scaling, rotating, moving and deleting (see Figure 7). The 3D volumes are rectangular with a texture indicating floor divisions. Each floor in a building volume can be manipulated by setting its height, and the 3D volumes show information about floor area and height. It is possible to choose the colors of the building volumes. Users can also select a base map that can be overlayed on the terrain model. 3D models and image files can be imported and placed on the 3D mesh model by scaling, rotating and moving. For now, only certain types of files can be imported to the system, including FBX for 3D and PNG for images. The MVP is open for further development to include, e.g., motion capture VR.

Pins can be placed as interest points, and adding comments facilitates remote communication with other stakeholders. If a participant would like to use 1:1 on-site or scaled MAR, AR markers can be placed in the virtual environment, and the scale of both the physical marker (i.e., how big the printed marker should be in reality) and the 3D model projection (e.g., 1:400 tabletop scale or 1:1 on-site scale) can be decided by the participant. Using smartphones, participants can bring a printed QR marker to the site for 1:1 scale projection (see Figure 11) and design collaboratively directly on site or use the marker for the tabletop version MAR with the corresponding scale as indicated by the placed marker (see Figure 12). All participants can manipulate all objects created by other participants using both desktop computers and smartphones. It is possible to put volumes in different folders to facilitate turning on and off a group of objects. The consequences of a design can be assessed through sunlight and shadow simulation.

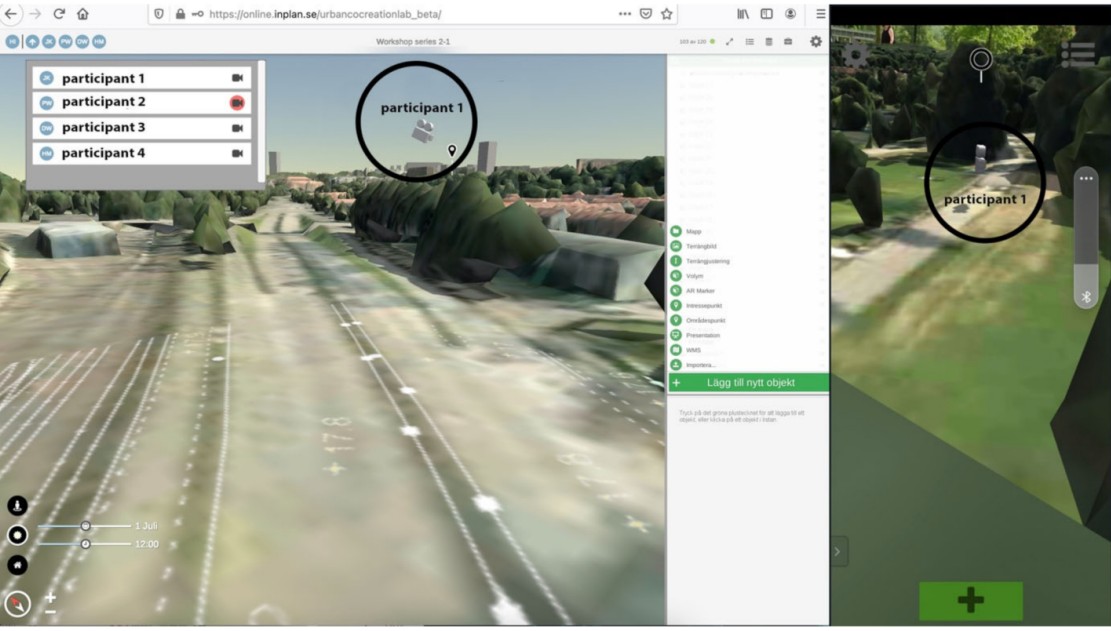

**Figure 8.** Participants can be seen in the form of a camera to show their location and their viewing direction. The web-based desktop version is seen to the left and MAR version to the right.

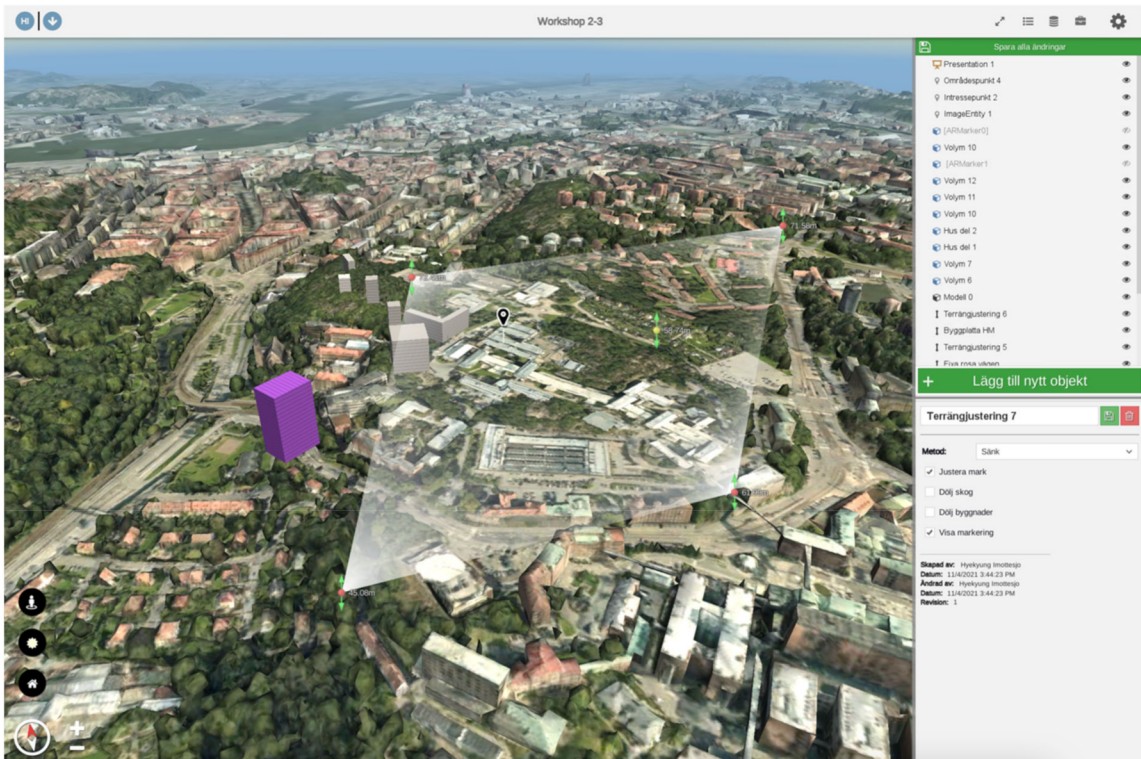

**Figure 9.** The terrain adjustment function, allowing users to change the terrain, including buildings.The information text for terrain adjustment has four check boxes: (from the top) Adjust terrain, Hide forest, Hide buildings, Show markings.

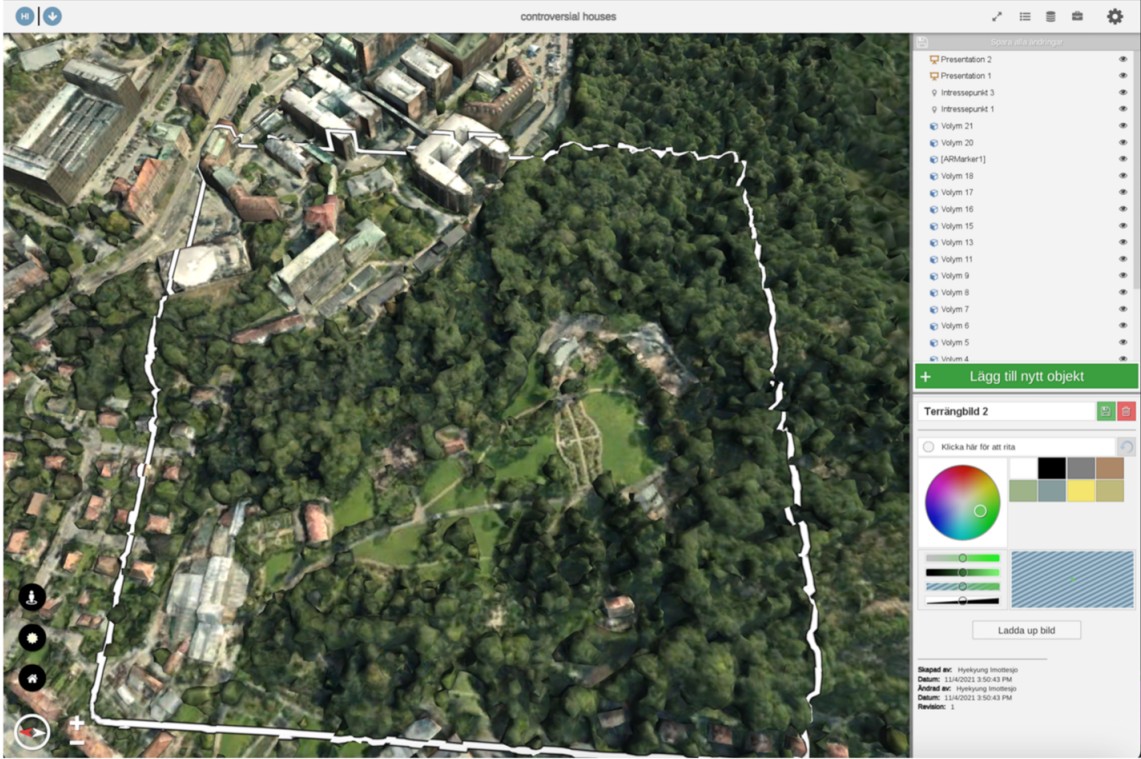

**Figure 10.** The terrain image function when placing objects, allowing users to draw 2D lines on the terrain.

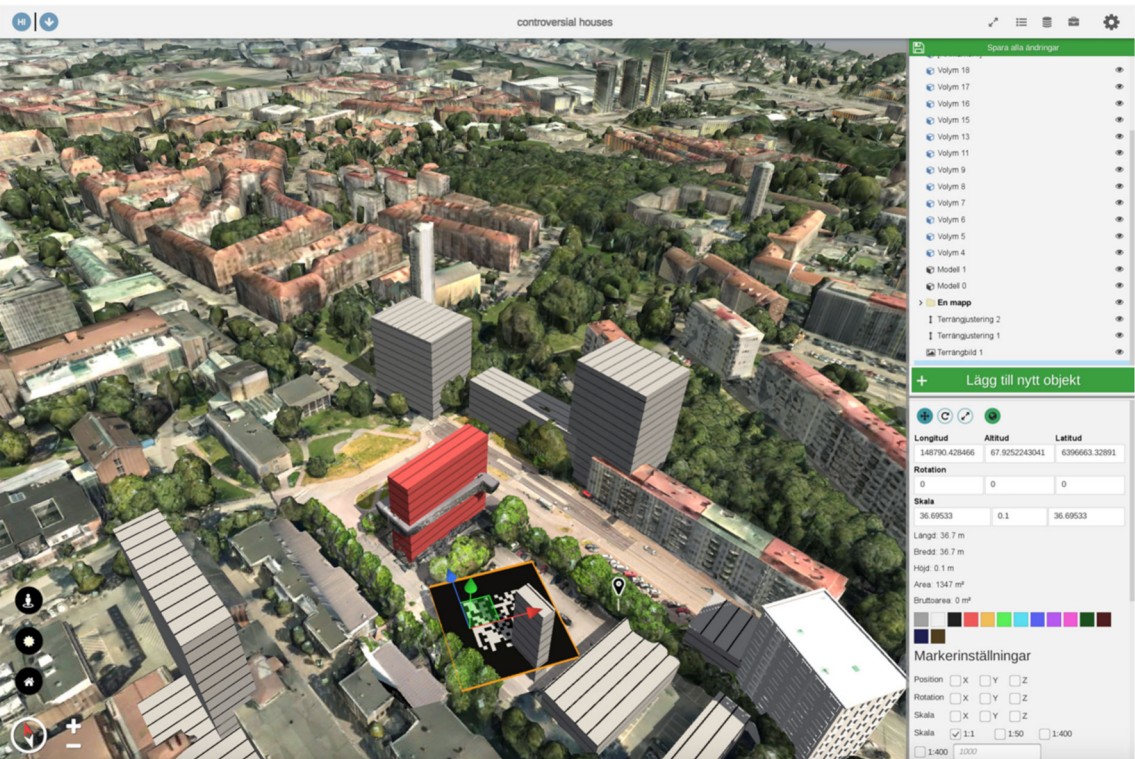

**Figure 11.** Using AR Markers when placing objects, allowing users to set the scale of projection and the size of the physical marker.

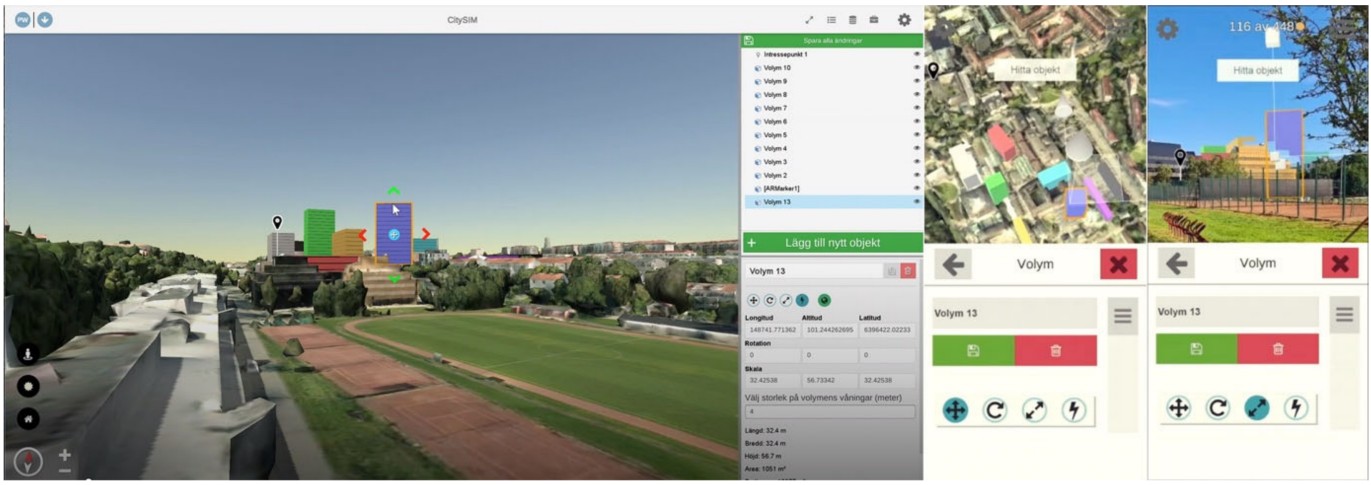

**Figure 12.** Showing synchronization between the web-based desktop version, scaled MAR and 1:1 on-site MAR.

**Table 2.** Functions of the MVP.

| Functions | | What Does It Do? |
| --- | --- | --- |
| Place objects | Volumes | A 3D cube is placed that can be moved, scaled and rotated. The dimension of the cube can be defined in an object property window. |
| | Pins | A pin with a variety of shapes can be placed. Information or comments can be attached by texts and photos. |
| | Terrain image | 2D lines can be drawn directly over the terrain. The color of the lines can be selected. An image file can be imported and overlaid on the terrain. The image file can be moved, scaled and rotated. |
| | AR markers | AR markers that can be moved, scaled and rotated are placed on the terrain. The scale of 3D projection can be determined in the object property window. The size of the physical marker can also be determined. |
| | Import | A 3D mesh model can be imported. The model can be moved, scaled, and rotated. |
| | Folders | Folders can be created and building volumes and 3D models can be placed in those. Folders can be turned on and off. |
| Terrain adjustment | | A part of the terrain can be defined by drawing a polygon over the location for adjustment. The nodes of the polygon can be elevated or lowered to adjust the elevation terrain. |
| Sun/shade simulation | | Sun and shade can be simulated for the design by manipulating the time of day and year. |
| Adopt the point of view of other participants | | Participants can adopt the point of view of other participants by choosing an avatar (camera icon with participant's name) to follow, including street-view and bird's-eye view. |
| Navigation | | One can navigate the scene in bird's-eye view and street-view. |

### 3.4. Elaborated Specifications as a Foundation for a Minimum Marketable Product

The two first workshops of Phase 2 resulted in a new list of issues and new wishes. These wishes were still quite generic and not detailed enough to develop an MMP. For example, there was a request to indicate building functions, but the stakeholders did not specify the what type of building functions or how to indicate them. Specific questions were therefore prepared for the final workshop in this phase. Table 3 accounts for the issues and wishes from the two first workshops of Phase 2 and the detailed discussion points developed by the researchers and software developers in preparation for the final third workshop. A summary of the ensuing workshop 3 discussions is also listed. During this workshop, the stakeholders were encouraged to discuss their wishes and issues in detail and consider if and how those wishes would need to be turned into specifications for the MMP. The final right-hand column shows the MMP specifications subsequently developed by the researchers and software developers. Issues that were discussed and considered "not essential to implement at this stage" were not turned into specifications for MMP at this stage.

**Table 3.** Sequence for developing detailed specifications for the MMP.

| Theme 1: Visualization | | | |
|---|---|---|---|
| **Scale** | | | |
| *Issues and Wishes* | *Discussion Points* | *Discussion Summary* | *Specifications for MMP* |
| Sense of scale is still an issue. | Are floor lines not enough in which way? | Floor lines and being able to incorporate the ground detail plan increases the perception of scale. | Not necessary to change. |
| | What else could be done for a better perception of scale? | Participants should be able to place urban objects, such as trees, cars, buses and benches, to understand the space better. | 1. Add library of urban objects, such as vegetation, vehicles, people and benches. |
| **Information** | | | |
| Building functions need to be indicated. | Does the function for changing the color of the building work for this? | Stakeholders need to be able to standardize colors for each building function. | 2. Allow user-defined standardization of colors for building functions. |
| | How would you indicate building functions? | Colors on different floors would increase the level of detail. | 3. Allow for applying colors on different floors for mixed function buildings. |
| More maps with different data should be available. | What additional types of maps and data would be available? | Availability of maps with data depends on the municipalities and what type of data they have. | |
| Create one's own map layer. | What type of map layers would need to be created? | Heatmaps with various information, roads, entrances to buildings and meeting places. | 4. Make it easy to create a map layer with user-defined information. |
| **Detailing** | | | |
| Better graphics are needed for a walkthrough. | The graphic quality depends on the 3D materials we can acquire for the project. What would be the minimum acceptable quality? | The 3D material provided by the municipality is not good enough to be useful for formal work processes. Project-based 3D environment mapping (e.g., through drones) is required. | 5. Possibility for project-based 3D environment mapping is required. |
| Different textures for the buildings, including transparency, might be needed. Same color building volumes make a wall-like effect viewed from a long distance. | Which textures would be good to include? Concrete, bricks, glass or any other? Should the users be able to upload textures to be used? | Textures would be great, but it is not necessary for basic usability at this stage. | |
| | Is transparency and reflection important? | Transparency reflection is not necessary at this stage. | |

**Table 3.** *Cont.*

| | | | |
|---|---|---|---|
| *Orientating and locating* | | | |
| Drop anchor points to go back to or send people to a specific location. | Is the "home button" not enough? | Users should be able to define locations to summon other participants. | 6. Users should be able to easily mark and unmark locations where to bring other users. |
| Eye-level remains on the original terrain level when walking on lowered ground level using a terrain adjustment function. | This is due to the type of 3D model provided by the municipality. | Distinguishing the vegetation and ground in the mesh model is required for better terrain modification and control of eye level. | 7. Possibility for project-based 3D environment mapping is required. |
| **Theme 2: Modeling** | | | |
| *Methods* | | | |
| Roads and 2D drawings from terrain adjustment function are drawn on top of vegetation. | This is due to the type of 3D model provided by the municipality. | | |
| Making building volumes by drawing polygons and extruding. | If polygons can be drawn and extruded for modeling, what would be good to have? Dimension, angles? | Simply choosing corners on the map and extruding would work. | 8. Making 3D volumes by choosing corners and extruding should be possible. |
| | | 2D drawing should enable vector lines that the users can manipulate using control points. | 9. 2D lines should be vector lines with control points. |
| More forms and shapes are needed. | What type of shapes are necessary? | Placing simple symbols such as arrows and circles. | 10. Include library of symbols, such as arrows and circles. |
| *Functionalities* | | | |
| Copy and paste function is necessary. | Any other essential functions? | Copy and paste of volumes. | 11. Allow for copy and paste of 3D volumes. |
| | | Copy and paste of terrain adjustment parameters. | 12. Allow for copy and paste of input parameters, such as terrain adjustment and building volumes. |
| *User interface* | | | |
| User interface | The text on the screen should be zoomed in and out. | Bug needs to be fixed. | |

**Table 3.** *Cont.*

| Theme 3: Remote simultaneous collaboration | | | |
|---|---|---|---|
| *Remote communication* | | | |
| Avatars should look different. | How much detail is needed for the avatar's look? Would color-coding work? | Avatars can be as they are with names of the participants indicated. | |
| Turning off a specific folder view with building volumes should be shared with other users to hide some of the building volumes for everyone. | Synchronizing the view for everyone? | Presentation mode where one can synchronize the views for everyone is required. | 13. Add presentation mode including a function to synchronize views for everyone. |
| Laser points to show where one person is pointing. | It is difficult to point at a specific object. How best can it be done? | A pointer, arrow or highlighting of a specific object should be available. | 14. Add indication for what is being pointed at. |
| Sessions should be recorded. | | | 15. Add a session recording function. |
| *Compatibility with other platforms* | | | |
| Importing PNG is problematic. | Which formats should be imported? | Starting with PNG of diverse sizes. | 16. Files of diverse sizes should be imported. |
| Importing Sketchup model is problematic. | | | 17. General improvement is required for importing and placing files. |
| Road maps need to be imported more easily. | | | |
| *Isolation and synchronizing* | | | |
| Feature to lock and unlock the volumes for everyone to prevent accidental removal of buildings. | | Users should be able to lock the volumes. | 18. Add function allowing users to lock volumes. |
| A visitor account to limit the number of people who can participate actively for a higher quality of discussion. | | This goes against the aim of the collaborative platform. However, this could be achieved already by inviting relevant stakeholders and using guest logging in. | |
| Theme 4: Simulation | | | |
| *Design consequences* | | | |
| Information related to sustainability, energy use and density could be simulated based on the design. | Can we create a specific list of relevant simulations? | Not necessary for this use. | |
| *Preset scenarios* | | | |
| Calculating the number of offices or apartments per building with pre-programmed values. | Can we create a specific list? | Not necessary at this stage. | |

## 4. Discussion

### 4.1. Visualizing the Built Environment

Understanding visual representations of a future built environment is critical for increasing the knowledgeability of stakeholders for their decision-making [21,27]. This study validates previous research, where immersion using mixed-reality solutions, such as AR and VR, has been found efficient for communicating a future built environment [18–21,28,32,34,37]. A photogrammetric mesh model of the city issued by the City of Gothenburg under an open data initiative was used as the basis for the virtual 3D environment. Even though the lightness of the data facilitates synchronization between devices, only depending on such a photogrammetric mesh model was not perceived as detailed enough by the stakeholders to properly understand the scale and to precisely locate objects in the model (see Section 3.4). Especially the vegetation appeared as broken polygons from the ground level on the desktop station, and this problem was even more prominent when it interfered with the AR projection due to rough edges appearing during occlusion (see Figure 13). Aerial laser terrain scanning integrated with the mesh model generated a more realistic 3D environment model [72] but was found to require a higher capacity to process data for the various devices integrated into the platform, such as smartphones.

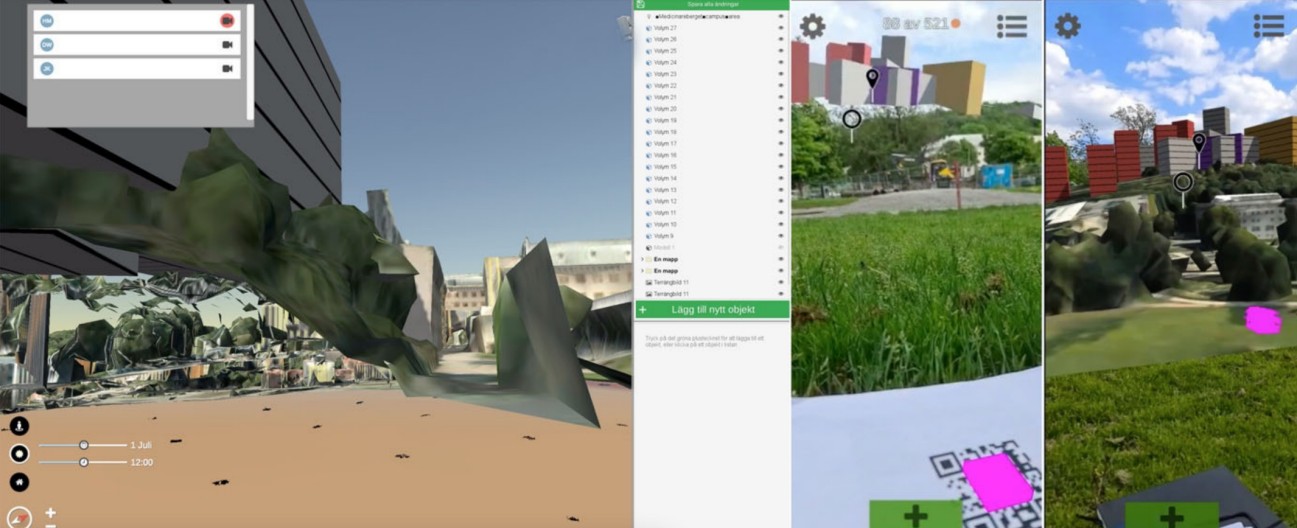

**Figure 13.** Problematic issues with the 3D mesh model. The vegetation is broken into polygons in close-up detail, and the building volumes are placed on top of the vegetation. The image to the left shows the web-based desktop station, the image in the middle MAR illustrations occlusion issues when the background 3D model is hidden, and the image to the right shows when the background mesh model is projected on-site. The roughness of the vegetation in the model is problematic when zoomed in. Likewise, the projection on MAR shows rough occluded cuts at the bottom (Image in the middle).

### 4.2. Layman Participation and Modeling Methods

In this study, the modeling method is simplified into the placement of a cube that can be manipulated. In previous MAR research, developing the first version of the Urban CoBuilder, the model was constructed by stacking cubes of $3 \times 3 \times 3$ m dimensions with different façade textures [37,70]. Users considered this method easy enough to start modeling quite complex and somewhat realistic buildings. However, the method required too many cube blocks to complete a building structure to be used as an urban planning tool, not least due to the processing capacity of the smartphones being too limited for managing such a high number of cube blocks [37,70]. In the present study, placing just one cube representing each building was lightweight enough for the processing capacity of various devices, but then the stakeholders were limited in creating the more elaborate

building forms typically needed in the recreation of planning projects. In response, various methods were proposed, such as drawing a polygon and extruding it to create the height of the building, but this needs to be tested further with a broader set of stakeholders. Such a method would also facilitate precision in building localization using MAR by participants walking from corner to corner of a proposed building on site, marking the building's shape and location. Although citizen involvement with support from a 3D modeler would probably benefit from more elaborate modeling methods, the current ambition for multiple stakeholders to create design input using their own devices requires further studies on how to balance simplicity (to satisfy usability and capacity of devices) and minimum necessities for shaping relevant urban forms.

### 4.3. Remote Collaboration

In this study of multiple stakeholder design collaboration, each participant created new forms and modified existing ones simultaneously. As the participants could modify all forms that exist within the virtual environment, the forms created by other stakeholders were also subjected to changes without the intent of the creator stakeholder. This was considered problematic, especially by architect and property developer stakeholders, since ownership of a design also entitles responsibility for that very design during a work process. Collaboration of this type would resemble a multi-user CAD project where users remotely and simultaneously can edit parts of a CAD model [73]. Such a project design emphasizes the need for communication between the users [74] to transfer the participants' perspectives among users to have a general view on what other participants are currently working on and, not least, to resolve issues deriving from invading other participants' "design spaces". Using a form of an avatar to indicate what other participants are currently engaged with can also help the flow of collaboration [73] while also increasing creativity [59].

The Urban CoCreation platform implemented both these strategies, i.e., the possibility to adopt other participants' points of view and utilize avatars as an aid for collaboration. However, the issues of locking a model or isolating and/or locking modifiable design features were considered critical for claiming ownership and limiting and controlling what each stakeholder can engage with. This aspect would be essential for collaboration among stakeholders engaged with specific tasks, such as building engineers and architects. Still, consideration needs to be taken when locking and assigning levels of control; the purpose of the platform is to be able to take in many and diverse stakeholders, generating as much input as possible. For citizen participation, even though having one's design deleted and changed by others spontaneously might be problematic, allowing each citizen to claim a model and lock it would also create problems. Careful design is needed of communication channels between the stakeholders, of possibilities to adopt the point of view of others, of avatars to clearly understand who is working on which item, of ways to mark creatorship of each design and of mechanisms to lock some of the models.

### 4.4. Simulation of Design Impact

The workshops uncovered a strong wish among stakeholders to be able to simulate design impacts related to sustainability issues, such as energy use and noise levels. Simulative tools are available for life cycle analysis (LCA) based on building designs, even though most such tools focus on the detailed design stages [75,76]. Even though some of the tools can be used in early phases of projects, such simulation might imply model imprecision due to "input uncertainty, and data variability" [76]. Still, Hollberg et al. (2021) argue that such tools should be available to building-design professionals through more accessible interfaces during the early design phase. Simplifying the building models could be one way of achieving this (for an example with LCA, see [77]), where the current MMP only uses color-codes for building materials and functions, without specifying any details regarding, e.g., exterior walls and windows. In this regard, to make the co-creation platform more useful, additional studies are needed to identify the required minimum input details related to building design and building data.

## 5. Conclusions

Developing a collaborative designing platform to include a diversity of stakeholders in urban development processes entails considering the variety of needs of the stakeholders arising from different interest areas and varying proficiency in using tools for visualization. The first series of workshops—testing the use and integration of web-based desktop 3D modeling, head-mounted VR and MAR applied to different scales, including 1:1 scale on-site—identified a diverse set of stakeholder wishes with relevance for the platform to be valid. Based on specifications developed from this wishlist, an MVP was developed and tested in the second series of workshops, involving property developers as stakeholders with a real campus development project as a test site. A new set of more detailed specifications for an MMP were developed based on a revised stakeholder wishlist. Even though the second set of stakeholder workshops mainly focused on testing the web-based desktop 3D modeling, the integration of MAR was still tested by the researchers and the software developers. The gathered insights identified challenges in need of further investigation, e.g., related to minimum required levels of detailing in environmental impact simulation and perception of scale.

Even with the study's limitations described below, the usefulness of the platform was confirmed. It showed a capacity for quick visualization of urban concepts, e.g., different heights and locations of buildings, and possibilities to view the impact of site development from other parts of the city where a changed sightline would be of significance. Through discussions based on street level walk-through and adopting each other's viewpoints, the developers discovered new potential connections to neighboring building blocks as well as problems inherent in some of the proposals by their architects, e.g., that the buildings aligned to create a wall when viewed from a central city district, blocking the sightlines.

### 5.1. Limitations

The second phase of the study focused on a particular set of stakeholders' work processes using the web-based desktop version of the platform. This led to three limitations potentially affecting the outcomes. The first concerns the limited stakeholder types involved, i.e., only those engaged in development projects at a larger real estate company with their specific set of requirements and competencies. Thus, the design of the user-interface related to the levels of expertise of diverse stakeholders was not studied extensively. Second, at this stage, the platform was only used in discussions oriented towards the very early phase of an urban development project, leading to certain types of topics being explored but excluding others. Third, COVID-19 limited the type of device tested with the stakeholders, i.e., only remote web-based desktop 3D modeling. Still, as the desktop version is integrated with the VR and AR interfaces, VR and MAR integration could be tested to some extent by the researchers and the software developers in support of the development of the MMP. Furthermore, limiting the scope of application of the platform made it possible to delve more deeply into the specifications needed to be fulfilled for it to be marketable and applicable, at least in a specific phase of urban development.

### 5.2. Scope for the Next Phases

The intention is to gradually broaden the scope of the platform by applying it in different phases of urban development processes and with different sets of stakeholders. Launching the platform as a usable MMP in real projects is expected to facilitate such testing in broader situations, and hopefully, less affected by COVID-19 restrictions to provide increasingly richer feedback. In the next phase of the project, additional stakeholder workshops including VR and MAR are planned. The main objective is to increase stakeholder diversity by including architects, city planners and neighbors of the campus development project and to include later phases of the urban development process.

Even though extensive further development is required to significantly increase the platform's scalability, the issue of scalability and extensibility will still be considered already in the next phase. An option of developing a project-based 3D drone mapped site model

that is compatible with the platform will be considered to support the scalability of the platform. In that case, project-based drone-mapped 3D modeling of the project site would involve additional studies regarding the minimum required level of detail for productive collaborative modeling and visualization while still fulfilling criteria for efficient mapping of the environment. In addition, a study on the perception of scale linked to the level of graphical detail and the stakeholder's pre-knowledge in virtual reality platform and 3D modeling would serve to improve the user interface, facilitating non-expert stakeholders input using the collaborative designing platform.

**Supplementary Materials:** The following supporting information can be downloaded at https://www.mdpi.com/article/10.3390/app12020797/s1, AGENDA for workshop series 1, Observation and discussion points workshop 2-1, Questionnaire workshop 2-1, Questionnaires and discussion points workshop series 1, Questions and discussion points for workshop 2-3.

**Author Contributions:** Conceptualization, H.I.; methodology, H.I., J.-H.K.; validation, H.I., J.-H.K.; formal analysis, H.I.; investigation, H.I, J.-H.K.; writing—original draft preparation, H.I.; writing—review and editing, H.I., J.-H.K.; visualization, H.I.; project administration, H.I.; funding acquisition, H.I. All authors have read and agreed to the published version of the manuscript.

**Funding:** The project is funded by Swedish government research council for sustainable development (FORMAS) and by the Adlerbertska Foundation.

**Conflicts of Interest:** The authors declare no conflict of interest.

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
