# Peer review of "The Urban CoCreation Lab—An Integrated Platform for Remote and Simultaneous Collaborative Urban Planning and Design through Web-Based Desktop 3D Modeling, Head-Mounted Virtual Reality and Mobile Augmented Reality: Prototyping a Minimum Viable Product and Developing Specifications for a Minimum Marketable Product"

_applsci, doi:10.3390/app12020797_

Round 1

Reviewer 1 Report

The manuscript appears like a very mature research manuscript reporting the design and application of a web-based 3D modeling tool used in participatory planning processes. There are only minor points which the authors could consider when preparing a revised version of the manuscript:

  • How does your application deal with ecological indicators of urban landscape planning, a topic getting more and more attention in sustainable urban development (see https://doi.org/10.1016/j.ecolind.2011.12.004)

  • In section 1.1.1, you mention (the importan aspect of) data transparency, without going into detail on that matter. The growing resources of open geospatial data however play an important role when developing VR- and AR-based media used in urban planning (see for e.g. https://doi.org/10.1007/s42489-020-00069-6 and https://doi.org/10.1007/978-3-319-47289-8_17 )

  • How did you adress the problem that different stakeholders (study participants) have different pre-knowledge in operating with (geospatial) data and the technical devices you used? Is there any empirical way to minimize the effects between experts and non-experts?

Author Response

Dear Reviewer 1,

Thank you for your engaged reading of our paper and the valuable comments. You highlight important issues, and this revision has significantly improved the paper, for which we are grateful.

Reviewer 2 Report

The paper has presented the interesting research and development process of an integrated platform named as Urban CoCreation Lab.  This platform aims to support multiple users with remote and simultaneous collaborative urban planning and design and combine the web-based desktop 3D modelling, VR and Mobile Augmented Reality technologies.  The paper was well-written with a clear structure.

Iterative prototyping method has been applied and this seems work well with such kind of project but it would be nice to justify why this method was chosen. 

The platform implementation is complex and the paper covers the MVP and MMP prototyping development and there are some discussions about the compatibility and information support but it will be nice to briefly discuss the extensibility and scalability of the platform and explain the potentials of applying this platform in other cases. 

Round 2

Reviewer 1 Report

The authors provided a revised version of the manuscript and a response letter addressing all (minor) points mentioned in the first review round. The argumentation in the response letter is clear. Therefore, I would like to recommend the present manuscript for publication in this journal.